# Bioinspired Garra Rufa Optimization-Assisted Deep Learning Model for Object Classification on Pedestrian Walkways

**DOI:** 10.3390/biomimetics8070541

**Published:** 2023-11-11

**Authors:** Eunmok Yang, K. Shankar, Sachin Kumar, Changho Seo

**Affiliations:** 1Department of Financial Information Security, Kookmin University, Seoul 02707, Republic of Korea; emyang@kookmin.ac.kr; 2Department of Computer Science and Engineering, Saveetha School of Engineering, Saveetha Institute of Medical and Technical Sciences, Chennai 602105, India; drkshankar@ieee.org; 3Big Data and Machine Learning Lab, South Ural State University, Chelyabinsk 454080, Russia; 4College of IBS, National University of Science and Technology, MISiS, Moscow 119049, Russia; kumar.s@misis.ru; 5Department of Convergence Science, Kongju National University, Gongju-si 32588, Chungcheongnam-do, Republic of Korea; 6Basic Science Research Institution, Kongju National University, Gongju-si 32588, Chungcheongnam-do, Republic of Korea

**Keywords:** bioinspired algorithms, image classification, object detection, deep learning, pedestrian walkways

## Abstract

Object detection in pedestrian walkways is a crucial area of research that is widely used to improve the safety of pedestrians. It is not only challenging but also a tedious process to manually examine the labeling of abnormal actions, owing to its broad applications in video surveillance systems and the larger number of videos captured. Thus, an automatic surveillance system that identifies the anomalies has become indispensable for computer vision (CV) researcher workers. The recent advancements in deep learning (DL) algorithms have attracted wide attention for CV processes such as object detection and object classification based on supervised learning that requires labels. The current research study designs the bioinspired Garra rufa optimization-assisted deep learning model for object classification (BGRODL-OC) technique on pedestrian walkways. The objective of the BGRODL-OC technique is to recognize the presence of pedestrians and objects in the surveillance video. To achieve this goal, the BGRODL-OC technique primarily applies the GhostNet feature extractors to produce a set of feature vectors. In addition to this, the BGRODL-OC technique makes use of the GRO algorithm for hyperparameter tuning process. Finally, the object classification is performed via the attention-based long short-term memory (ALSTM) network. A wide range of experimental analysis was conducted to validate the superior performance of the BGRODL-OC technique. The experimental values established the superior performance of the BGRODL-OC algorithm over other existing approaches.

## 1. Introduction

Recent technological developments such as computer vision (CV) and surveillance cameras (CCTV) have been utilized to protect the pedestrians and support safer walking practices. For this purpose, the type of characteristics of risk constituents is required to save the pedestrians from accidents [1]. Numerous CV techniques have been developed by focusing on the processes such as activity learning, feature extraction, data acquisition, behavioral learning, and scene learning. The main objective of such techniques is to compute the operations, such as video processing systems, traffic monitoring, scene identification, multicamera-relied challenges and methods, human behavior learning, activities analysis, vehicle prediction and monitoring, anomaly prediction techniques, etc. [2]. The current study focuses on anomalous forecast, a subfield of behavioral learning from the captured visual images. Moreover, anomalous prediction methods realize popular behavior with the help of training processes. The presence of numerous significant variations in the standard implementation process is defined as “anomalous” [3]. Specific instances of anomalies include cross-walking, presence of vehicles on paths, collapse of individuals while walking, signal avoidance at traffic junctions, vehicles making U-turns in red signals, and the unpredicted allocation of people in the crowd [4].

Pedestrian detection involves the automated identification of the walking persons from the information gathered from video and image sequences as well as the accurate location of the pedestrian region [5]. However, pedestrians can be identified as nonrigid objects in difficult environments, in various positions, under varying light conditions, and with changing levels of occlusion in real road situations. These scenarios increase the difficulty of accurately identifying the pedestrians [6]. With the fast growth of artificial intelligence (AI) technology, pedestrian identification has become a significant research area in CV. Pedestrian identification research approaches have been commonly categorized into two types, namely, conventional and deep learning (DL)-based identification techniques [7].

The DL technique is an advanced domain in the machine learning (ML) field that aims to determine the complex models of modest representations. The DL algorithms commonly depend on artificial neural networks (ANNs) that contain numerous hidden layers with nonlinear processing components [8]. The term “deep” corresponds to the presence of several hidden layers that are employed to modify the representation of the data. By applying the idea of feature learning, all the hidden layers of the neural networks design their input data in a new representation [9]. The layer control engages a higher level of generalization than the theoretical perception in the preceding layer. In DL frameworks, the hierarchy of feature learning at numerous levels is ultimately mapped to the output of the ML technique in one architecture [10]. Like ML algorithms, the DL approach can be categorized into two different classes such as unsupervised learning methods and supervised learning techniques, comprising deep neural networks (DNNs).

The current research paper outlines the design of the bioinspired Garra rufa optimization-assisted deep learning model for object classification (BGRODL-OC) technique on pedestrian walkways. The BGRODL-OC technique primarily applies the GhostNet feature extractor to produce a set of feature vectors. Moreover, the BGRODL-OC technique makes use of the GRO algorithm in the hyperparameter tuning process. Finally, the object classification process is performed using the attention-based long short-term memory (ALSTM) network. A wide range of experimental analysis was conducted to validate the superior performance of the BGRODL-OC method. In short, the key contributions of the paper are summarized herewith. 

An effective BGRODL-OC technique is developed in this study, comprising GhostNet feature extraction, GRO-based hyperparameter tuning, and ALSTM-based classification for pedestrian walkway detection. To the best of the authors’ knowledge, the BRGODL-OC technique has never been mentioned in the literature.The GhostNet model is developed to produce a collection of feature vectors. This model is known for its efficiency and effectiveness in deep-learning-based image analysis and in improving the accuracy of object detection.The BRGO algorithm is employed for the hyperparameter tuning process, which helps in fine-tuning the model’s parameters to improve its performance in object classification.The ALSTM model is presented for the object classification process, which can capture long-term dependencies in video data. The attention mechanism enhances the model’s ability to focus on relevant information, thus further improving the accuracy.

## 2. Related Works

Abdullah and Jalal [11] presented a new technique using the DL framework and conditional random field (CRF). In this study, the preprocessing was executed primarily, while the superpixels were produced secondarily, utilizing enhanced watershed transform. Then, the objects were segmented using a CRF. The relevant field was localized by employing the conditional probability while a temporal relationship was applied to find the areas. At last, a DL-based hierarchical network method was exploited for identification and classification of the objects. In [12], the authors proposed the automatic DL-based anomaly detection technology in pedestrian walkways (DLADT-PW) technique for susceptible transport user’s protection. The suggested technique comprised preprocessing as the main phase to be implemented for eliminating the noise and increasing the quality of the image. Similarly, the mask region convolutional neural network (Mask-RCNN) with DenseNet technique was utilized for identifying the operations. Harrou et al. [13] developed an innovative deep hybrid learning approach with a completely-directed attention module. The presented technique increased the modeling ability of the variational autoencoder (VAE) by combining it with the LSTM algorithm and employing a self-attention module at multiple phases of the VAE method.

Al Sulaie [14] introduced a novel golden jackal optimizer with DL-based anomaly detection in pedestrian walkways (GJODL-ADPW). In the developed GJODL-ADPW method, the Xception technique was utilized for efficient extraction of the feature method. The GJO technique was employed for optimal selection of the hyperparameters. Lastly, the bidirectional-LSTM (Bi-LSTM) methodology was implemented with an aim to detect the anomalies. Jayasuriya [15] suggested a method with the help of the convolutional neural network (CNN) approach. In this study, the localization was performed on a predesigned map. The extended Kalman filter (EKF) combines such annotations. In addition to this, an omnidirectional camera was added to the technique to enhance the efficient field of view (FoV) of the landmark detection method. The data-theoretic approach was also exploited to select a better viewpoint. Alia et al. [16] designed a hybrid DL technique along with a visualization model. This architecture had two key mechanisms. Firstly, deep optical flow and wheel visualization were utilized to produce the motion data maps; secondly, a false reduction method and EfficientNet-B0-based classifier were incorporated.

Kolluri and Das [17] implemented a technique by employing the hybrid metaheuristic optimizer with DL (IPDC-HMODL) in which 3-phase was offered. Primarily, the IPDC-HMODL approach employed multiple modal object detectors through RetinaNet and YOLO-v5 frameworks. Secondarily, the IPDC-HMODL technique implemented the kernel extreme learning machine (KELM) method for the classification of the pedestrians. Lastly, the hybrid salp swarm optimization (HSSO) method was utilized to optimally adapt the parameters. Alsolai et al. [18] introduced the innovative sine cosine algorithm with DL-based anomaly detection in pedestrian walkways (SCADL-ADPW) technique. This approach employed the VGG-16 framework for producing the feature vectors. In addition, the SCA algorithm was developed for the optimal hyperparameter tuning methods. In this study, the LSTM approach was exploited for anomaly detection.

## 3. The Proposed Model

In the current manuscript, an automatic object classification technique on pedestrian walkways termed BGRODL-OC method is developed. The objective of the BGRODL-OC algorithm is to recognize the presence of pedestrians and objects in the surveillance video. It encompasses several processes such as the GhostNet feature extractor, the GRO-based hyperparameter selection, and the ALSTM-based classification. Figure 1 illustrates the workflow of the BGRODL-OC algorithm.

### 3.1. Feature Extraction: GhostNet Model

In this phase, the GhostNet model is applied for the feature extraction process. The fundamental breakthrough of the GhostNet model is the introduction of the Ghost modules that reduce the number of convolutional sizes and calculations through cheap linear transformation so as to generate redundant feature mapping [19]. It also uses initial and cheap convolutions instead of typical convolutions. The input dataset is $X\in R\left(C\times H\times W\right)$, in which the H, W, and C correspond to the number of height, width, and channels, and X first passes over the convolution kernels to be 1×1 first convolutions once the network is trained.
(1)$$Y^{\prime }=X\times F_{1\times 1}$$

Here, $F_{1\times 1}$ refers to pointwise convolution and $Y^{\prime }\in R\left(H\times W\times C_{out}^{\prime }\right)$ shows the inherent features. Next, the cheap convolution is used to generate further features and interconnect the generated features through the first convolution, as given below.
(2)$$Y=Concat\left(\left[Y^{\prime },Y^{\prime }\times F_{dp}\right]\right)$$

In Equation (2), $F_{dp}$ refers to depthwise convolution and $Y\in R(H\times W\times C_{out})$ indicates the output features.

Though the GhostNet model reduces the computational cost, its ability to capture the spatial information is reduced. Thus, the Ghostnet-V2 model adds another attention mechanism, i.e., DFC, based on the FC layer that has low hardware requirements. It is achieved by capturing the dependencies amongst longer distance pixels, and it can enhance the inference speeds. The computation of DFC attention is as follows. 

Assume $\in R\left(H\times W\times C\right)$ as an HW label, $z_{i}\in R^{C},Z=\left\{z_{11},z_{12},\ldots ,z_{HW}\right\}$. The features are aggregated along the vertical and horizontal directions, correspondingly, and are formulated using the following equations.
(3)$$\alpha_{hw}^{\prime }=\sum_{h^{\prime }=1}^{H}F_{h,h^{\prime }w}^{H}⊙z_{h^{\prime }w^{\prime }},h=1,2\cdots H,w=1,2,\cdots W,$$
(4)$$\alpha_{hw}=\sum_{w^{\prime }=1}^{\Sigma y}F_{w,hw^{\prime }}^{W}⊙a_{h^{\prime }w^{\prime }}^{\prime },h=1,2\cdots ,H,w=1,2,\cdots ,W,$$
Here, $F^{H}$ and $F^{W}$ denote the transformation weights, ⊙ indicates the elementwise multiplication, and $A=\left\{a_{11},\;a_{12},\;\ldots ,\;a_{HW}\right\}$ represents the generated attention map. The original feature Z is taken as input while the long-range dependency along both the directions is captured.

Equations (3) and (4) are the representations of DFC attention that aggregate the pixels in 2D horizontal and vertical directions, correspondingly. These equations partially exploit the shared transformation of weights and perform them with convolution to increase the inference speed. This also avoids the time-consuming tensor operations. Two depthwise convolutions, sized $1\times K_{H}$ and $K_{W}\times 1$, are used, independent of the feature map sizes to adapt the input images of various resolutions.

### 3.2. Hyperparameter Tuning: GRO Algorithm

The current study uses the GRO algorithm to adjust the hyperparameters related to the GhostNet architecture. The GRO algorithm is a procedure that employs the mathematical rules and is used to identify the better approach so as to determine the solutions for the problem [20]. The procedure starts by determining a main function that is normally connected to many engineering problems. Then, a group of parameters is defined and the constraints are overcome to attain the required outcomes. Once these are determined, the software then begins the optimization procedure that employs the mathematical models to identify the most efficient and effective parameter values for resolving this issue. The optimizer procedure is iterative, i.e., modifying the distribution of resources increases the performance. The GRO technique is performed in three parts: the GRO initialization, leader crossover, and the follower crossover.

**Procedure** **1.**
*GRO initialization*


The basic theory of GRO is to split the particles into several groups; each of the groups takes a unique group of leaders for either global or the local optimum group places. The GRO system also needs to deploy major rules like the notion that all the fishes can act as followers while the leaders rely on the connected global optimum point for all the groups. The follower portions can switch the very weak ones to stronger leaders, who can accomplish a better ideal value before the next iteration. It is essential to primarily adopt these maximal follower portions as a percentage. Further, the primary parameters are considered as the acceleration coefficients (cl, c2) and the inertia weighted (ω).
(5)$$followers\;number=\frac{total\;number\;of\;particles-number\;of\;groups}{number\;of\;groups}$$

**Procedure** **2.**
*Leaders’ crossover of GRO*


The GRO approach contains two leader crossover procedures that need to be considered in the study. A primary model involves the selection of the novel leaders for all the groups, whereas the secondary model involves the election of a great leader who can lead the maximal number of followers. These stages assist as the guiding rules that found the approaches to be a vital element, thus offering flexibility to this method.

**Procedure** **3.**
*Followers’ crossover of GRO*


There is a huge probability of determining the optimum performance from a problem space owing to the flexible motion among the groups. Some optimizer approaches are not as flexible to work from one searching space to another, which can cause confusion in most of the difficult issues. This problem appears because of the presence of numerous parameters and higher differential equation order from difficult problems. The GRO model deploys a process to seek a large space of the problem by utilizing the follower crossovers among the groups. First, an arbitrarily chosen fish, in all the groups, changes to a strong leader. Second, one step needs to be taken from the direction of all the leaders by evaluating the position (X) and velocity (v), employed in Equations (6) and (7), correspondingly.
(6)$$v_{i}\left(z+1\right)=\omega v_{i}\left(z\right)+c_{1}r_{1}\left(p_{i}\left(z\right)-X_{i}\left(z\right)\right)+c_{2}r_{2}\left(G_{i}\left(z\right)-X_{i}\left(z\right)\right)$$
(7)$$X_{i}\left(z+1\right)=X_{i}\left(z\right)+v_{i}\left(z+1\right)$$

The fitness function (FF) in the group figures is recomputed, containing every follower and leader. Equations (8) and (9) define a novel phase in the GRO method.
(8)$$moving\;followers_{i}=integer\left(E\times random\right)$$
(9)$$followers_{ij}=Max\left(\left(followers_{ij-1}-moving\;followers_{i}\right),0\right)$$
Here, f refers to the maximal feasibility of the moving fish. The pseudocode of GRO algorithm is given in Algorithm 1.
**Algorithm 1:** Pseudocode of the GRO algorithm   Select the primary values (number of particles, leader number, FF limits)   Followers number =n/leaders number   Compute FF for n of particles, with sort FF   While t< iteration do   For i=1 to leader counts   Upgrade particles for the follower for leaders(i) utilizing optimizer system    End for    i=2 to leader counts   Random£x= mobile_fishes(i)   Followers(i)=Max(0,followers(i)− mobile_fishes(i))   The total amount of mobile_fishes = total no. mobile_fishes+ mobile_fishes(i)   End for   Followers(1) = total no. of mobile_fishes + Followers(1)   Define the performance of sub-global for all the leaders   Compute the global solutionEnd while

In the GRO approach, fitness selection is a vital factor. The encoded solution is applied to measure the goodness of the solution candidate. Here, the accuracy values remain the primary condition to design an FF.
(10)$$Fitness=max\left(P\right)$$
(11)$$P=\frac{TP}{TP+FP}$$
Here, TP and FP represent the true and false positive values, respectively.

### 3.3. Object Classification: ALSTM Model

The ALSTM model is used for the object classification process in this study. The underlying concept of the LSTM network is to control the data flow through gates and to utilize the memory cells (units) for storing and transferring the data [21,22]. Particularly, the LSTM network includes a memory cell along with input, forget, and output gates.

The input gate defines the amount of data that are fed as input to the memory units while the forget gate decides whether to remove the prior memory or not. Finally, the output gate decides the output of the hidden layer. The memory units are accountable for storing and transmitting long-term data that can be updated and controlled by calculating the gate units. Now, $x_{t}$ refers to the input dataset at t time${;\;C}_{t-1}$ denotes the memory values at t − 1 time; $h_{t-1}$ indicates the output values of the LSTM network at t −1 time. The three datasets $x_{t},$ $C_{t-1}$, and $h_{t-1}$ constitute the input dataset. $C_{t}$ corresponds to the memory values at t time, $h_{t}$ represents the output values of the LSTM network at t time, and the two datasets $C_{t}$ and $h_{t}$ constitute the output information.

The control functions of the forget, input, and the output gates are as follows.
(12)$$f_{i}^{\left(t\right)}=sigma\left(b_{i}^{f}+\sum_{j}U_{i,j}^{f}x_{j}^{\left(t\right)}+\sum_{j}W_{i,j}^{f}h_{j}^{\left(t-1\right)}\right)$$
(13)$$g_{i}^{\left(t\right)}=sigma\left(b_{i}^{g}+\sum_{j}U_{i,j}^{g}x_{j}^{\left(t\right)}+\sum_{j}W_{i,j}^{g}h_{j}^{\left(t-1\right)}\right)$$
(14)$$q_{i}^{\left(t\right)}=sigma\left(b_{i}^{0}+\sum_{j}U_{i,j}^{0}x_{j}^{\left(t\right)}+\sum_{j}W_{i,j}^{0}h_{j}^{\left(t-1\right)}\right)$$
where $b_{o},\;Uc,\;\mathrm{and}\;Wo$ refer to the bias, the input, and the cyclic weights of the forget gate, correspondingly.

Attention mechanism is the important component used in the NN model. The core principle is to allocate the attention weight to dissimilar parts of the input datasets, thus reducing the role of inappropriate parts. At the time of processing and learning tasks, this allows one to be more focused on the crucial data, which eventually improves the performance. This attention weight is used for computing the context vectors that capture the fittest data from the inputs.

In order to improve the model’s performance, the relevant equation is given below.
(15)$$\alpha_{i}=\frac{e^{\left(s\left(h_{i},h_{t}\right)\right)}}{\Sigma_{j=1}^{N}e^{\left(s\left(h_{i},h_{t}\right)\right)}}$$
(16)$$\alpha =\sum_{i=1}^{N}\alpha_{i}h_{i}$$

Now $\alpha_{j}$ denotes the score of the feature vector and a high score designates great attention. $s(h_{i},h_{t})$ shows the weight value of the ith input feature in the attention module, viz., the score ratio of the feature vector to the entire population. Next, each vector is added and averaged to attain the concluding vector, α. Figure 2 defines the architecture of the ALSTM network.

## 4. Results and Discussion

The proposed model was simulated in Python 3.6.5 tool on a PC configured with specifications such as i5-8600k, GeForce 1050Ti 4 GB, 16 GB RAM, 250 GB SSD, and 1 TB HDD. The parameter settings used for the study were as follows: learning rate: 0.01, dropout: 0.5, batch size: 5, epoch count: 50, and activation: ReLU. In this section, the performance of the BGRODL-OC technique is evaluated using the UCSD dataset [23], comprising images from the surveillance videos. Figure 3 depicts the sample images.

Table 1 shows the comparative accuracy ($accu_{y})$ examination outcomes achieved by the BGRODL-OC technique on test 004 and 007 datasets [12,24,25]. The results infer that the MDT and FRCNN models attained ineffectual performance. Moreover, the CBODL-RPD, DLADT-PW, and RS-CNN methodologies exhibited significant performance. However, the BGRODL-OC technique accomplished the maximum performance on all the frames.

Table 2 shows the average $accu_{y}$ analysis outcomes accomplished by the BGRODL-OC technique and other recent models on two datasets. Figure 4 portrays the comparative average $accu_{y}$ analysis results of the BGRODL-OC method with the existing systems on the test-004 dataset. The experimental values denote that both FR-CNN and MDT systems reached the least average $accu_{y}$ values such as 87.42% and 85.17%, respectively. In addition, the DLADT-PW and RS-CNN techniques achieved a moderate performance with average $accu_{y}$ values such as 98.35% and 97.90%, respectively. Although the CBODL-RPD model attained a considerable $accu_{y}$ of 99.06%, the BGRODL-OC technique exhibited the maximum performance with an average $accu_{y}$ of 99.32%.

Figure 5 shows the comparative average $accu_{y}$ analysis outcomes of the BGRODL-OC technique with present methods on the test-007 dataset. The experimental values specify that the FR-CNN and MDT techniques accomplished the least average $accu_{y}$ values such as 80.51% and 74.61% individually. Also, the DLADT-PW and RS-CNN methodologies achieved a modest performance with average $accu_{y}$ values such as 89.50% and 84.78%, correspondingly. While the CBODL-RPD algorithm simulated the data with a significant $accu_{y}$ of 92.27%, the BGRODL-OC system attained the maximum performance with an average $accu_{y}$ of 93.18%.

Table 3 and Figure 6 portray the comparative TPR results of the BGRODL-OC approach on test sequence 004. The results show that the DLADT-PW and MDT models obtained poor performance. Then, the CBODL-RPD technique reported slightly decreased performance. Simultaneously, the RS-CNN and FR-CNN methods accomplished considerable results. However, the BGRODL-OC technique outperformed other models with the maximum TPR values.

Table 4 and Figure 7 portray the comparative TPR analysis outcomes of the BGRODL-OC system on test sequence 007. The observational data denote that the DLADT-PW and MDT algorithms acquired inferior performance. In addition, the CBODL-RPD approach achieved a moderately low performance. Concurrently, the RS-CNN and FR-CNN methodologies attained notable experimental outcomes. However, the BGRODL-OC system outperformed the rest of the techniques with better TPR values.

Table 5 shows the comparative area under the ROC (AUC) curve and computation time (CT) results of the BGRODL-OC technique. Figure 8 shows the comparative AUC score results achieved by the BGRODL-OC technique. The results infer that the DLADT-PW, FR-CNN, RS-CNN, and MDT systems exhibited the worst performance, with the lowest AUC scores, such as 89.24%, 89.88%, 90.03%, and 89.28%, respectively. Though the CBODL-RPD technique attained a slightly enhanced performance with an AUC score of 96.54%, the BGRODL-OC technique surpassed the compared methods by achieving a maximum AUC score of 97.80%.

Figure 9 shows the comparative CT outcomes achieved by the BGRODL-OC technique and other recent approaches. The results imply that the RS-CNN and MDT algorithms achieved ineffectual outcomes, with maximum CT values such as 3.19 s and 3.56 s, respectively. At the same time, the CBODL-RPD, DLADT-PW, and FR-CNN methods accomplished moderate performance, with CT values such as 2.90 s, 2.75 s, and 2.90 s. However, the BGRODL-OC technique achieved an effectual performance with a minimal CT of 1.08 s. These results show the enhanced performance of the BGRODL-OC technique.

## 5. Conclusions

In the current study, an automatic object classification technique for pedestrian walkways termed the BGRODL-OC technique was developed. The objective of the BGRODL-OC technique is to recognize the presence of pedestrians and objects in the surveillance video. It encompasses several processes such as the GhostNet feature extractor, GRO-based hyperparameter selection, and the ALSTM-based classification. To achieve the objective, the BGRODL-OC technique primarily applies the GhostNet feature extractors to produce a set of feature vectors. In addition, the BGRODL-OC technique makes use of the GRO algorithm for hyperparameter tuning. Finally, the object classification is performed using the ALSTM network. A wide range of experimental analysis was conducted to validate the superior performance of the BGRODL-OC algorithm. The experimental values exhibited the better performance of the BGRODL-OC approach over other current techniques, with a maximum AUC score of 97.80%. The future works of the BGRODL-OC technique can enhance its scalability for handling real-time video streams and extend its applicability to different environmental conditions and camera perspectives so as to further bolster the pedestrian safety and object classification accuracy.

## Figures and Tables

**Figure 1 biomimetics-08-00541-f001:**
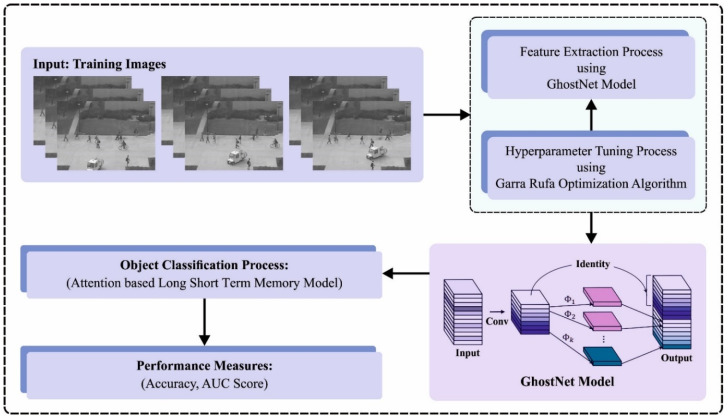
Workflow of the BGRODL-OC algorithm.

**Figure 2 biomimetics-08-00541-f002:**
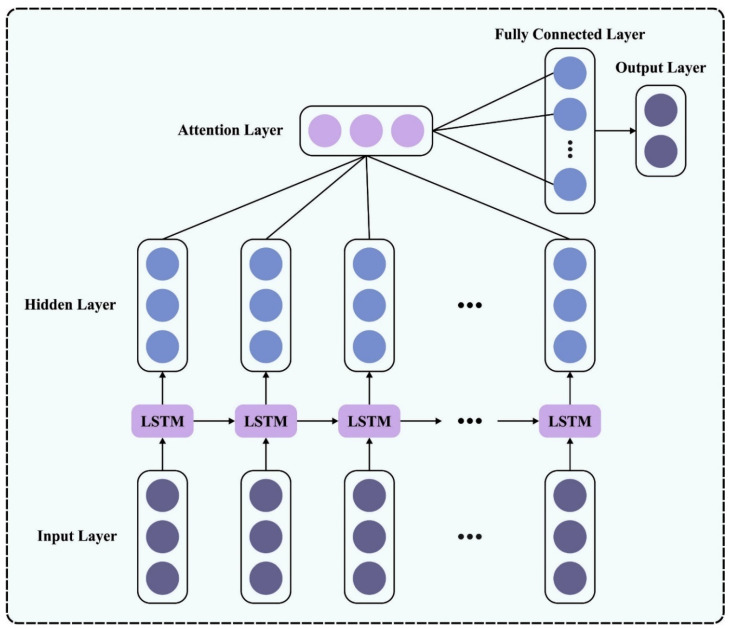
Architecture of the ALSTM.

**Figure 3 biomimetics-08-00541-f003:**
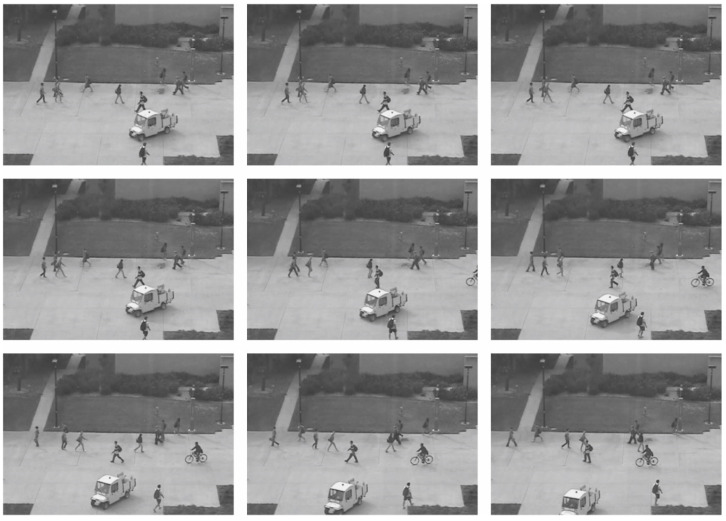
Sample images.

**Figure 4 biomimetics-08-00541-f004:**
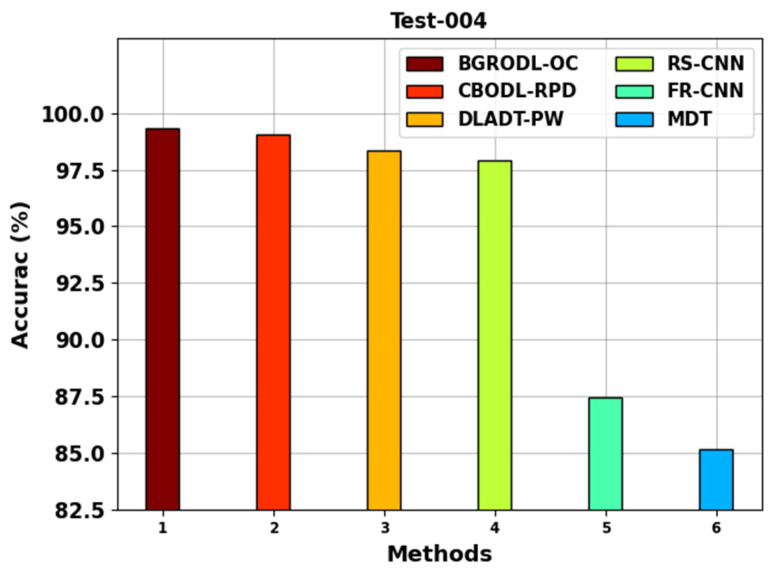
Average $accu_{y}$ analysis results of the BGRODL-OC approach on test 004 dataset.

**Figure 5 biomimetics-08-00541-f005:**
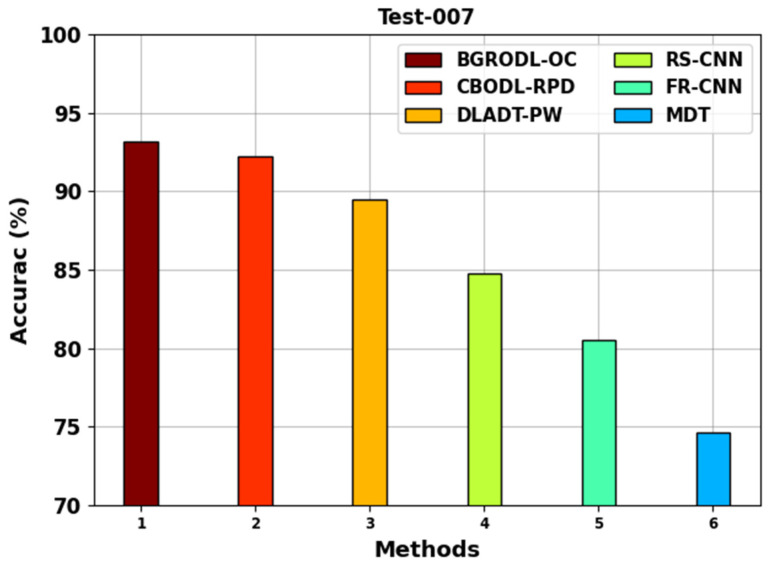
Average $accu_{y}$ analysis outcomes of the BGRODL-OC approach on test 007 dataset.

**Figure 6 biomimetics-08-00541-f006:**
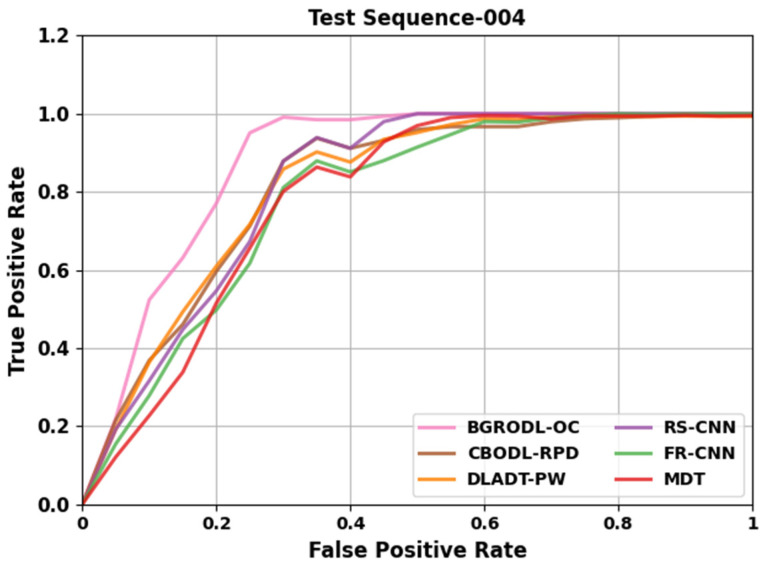
Comparative TPR outcomes of the BGRODL-OC technique on test sequence 004.

**Figure 7 biomimetics-08-00541-f007:**
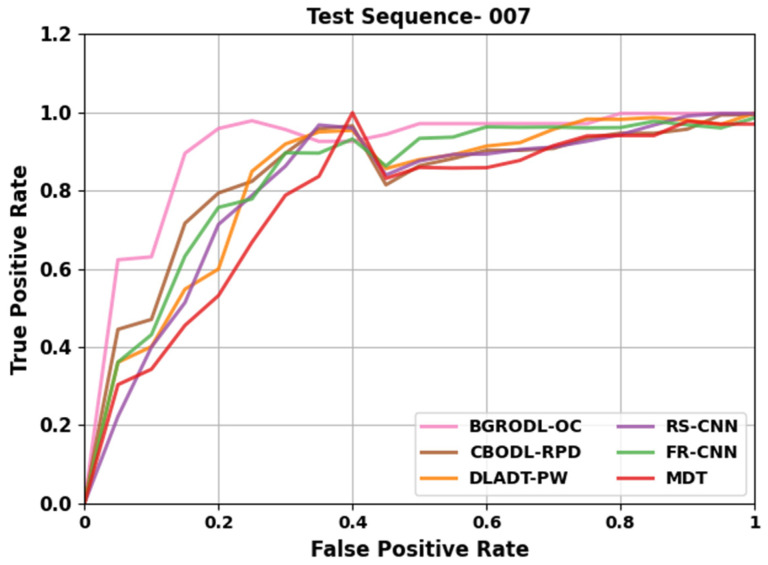
Comparative TPR outcomes of the BGRODL-OC technique on test sequence 007.

**Figure 8 biomimetics-08-00541-f008:**
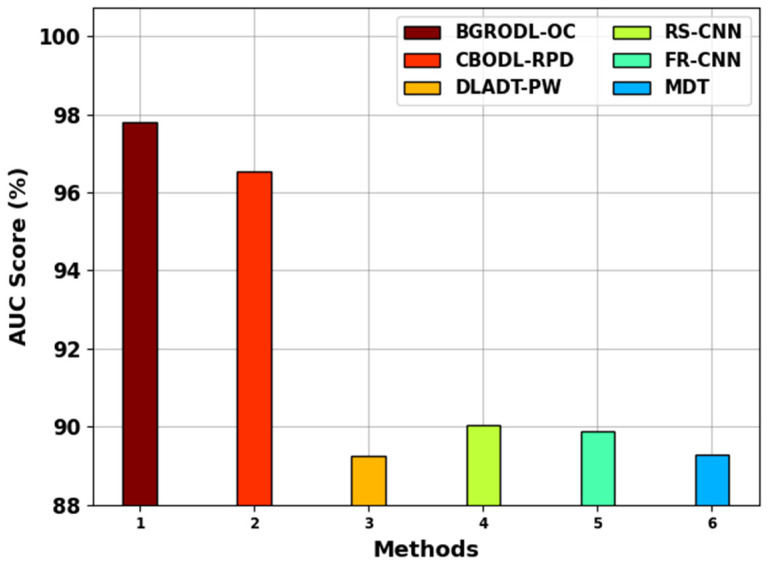
AUC score analysis outcomes of the BGRODL-OC technique and other methods.

**Figure 9 biomimetics-08-00541-f009:**
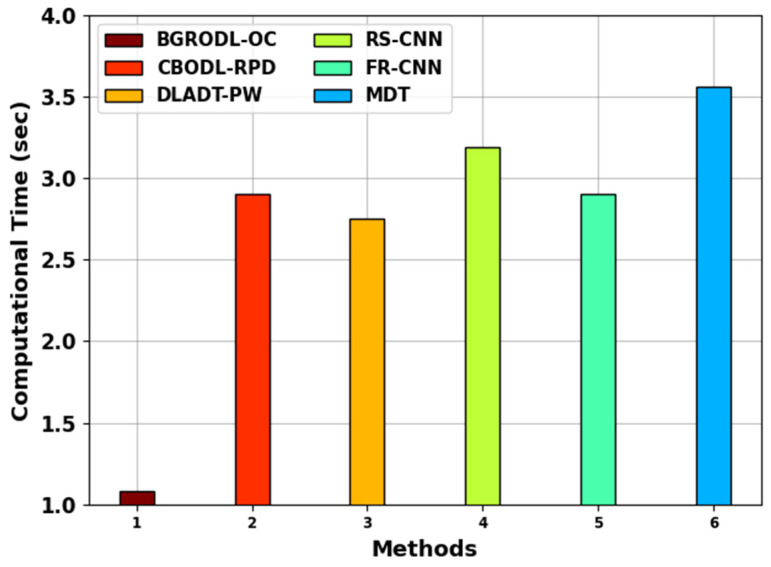
CT analysis outcomes of the BGRODL-OC technique and other techniques.

**Table 1 biomimetics-08-00541-t001:** $Accu_{y}$ analysis outcomes of the BGRODL-OC approach on test 004 and 007 datasets.

Test-004 Accuracy						
No. of Frames	BGRODL-OC	CBODL-RPD	DLADT-PW	RS-CNN	FR-CNN	MDT
FR-40	97.62	98.99	98.41	97.84	86.02	86.89
FR-42	99.45	98.98	98.23	97.65	88.90	85.67
FR-46	99.50	98.85	98.24	97.66	88.26	83.42
FR-51	99.38	99.61	99.32	97.60	85.88	86.39
FR-75	99.37	99.95	99.06	97.63	88.61	86.02
FR-106	99.90	98.96	98.07	98.27	88.92	86.99
FR-123	99.45	98.98	98.20	97.93	86.29	82.91
FR-135	98.93	98.93	98.06	97.89	86.25	83.26
FR-136	99.60	98.93	98.01	98.19	88.96	86.73
FR-137	99.07	98.91	98.08	98.25	87.34	86.13
FR-149	99.82	98.90	98.04	98.10	86.00	84.86
FR-158	99.85	98.99	98.20	98.04	88.51	86.59
FR-177	98.91	98.94	98.20	97.67	86.48	84.51
FR-178	99.51	99.00	99.09	97.78	88.79	84.05
FR-180	99.40	98.98	98.07	97.95	86.05	83.20
**Test-007 Accuracy**						
**No. of Frames**	**BGRODL-OC**	**CBODL-RPD**	**DLADT-PW**	**RS-CNN**	**FR-CNN**	**MDT**
FR-78	98.55	97.32	97.33	92.56	86.74	78.77
FR-91	100.64	100.91	98.01	93.97	87.70	77.38
FR-92	100.45	100.16	100.68	95.52	88.72	67.78
FR-110	100.49	99.57	97.05	93.86	85.92	73.46
FR-113	97.60	96.11	94.90	90.33	86.89	76.72
FR-115	89.41	88.27	85.81	85.24	84.60	73.46
FR-125	100.44	100.31	99.80	94.13	91.76	71.02
FR-142	100.06	99.44	99.49	97.12	84.40	71.34
FR-146	87.11	85.98	86.62	82.64	77.18	81.28
FR-147	90.71	89.41	85.68	83.85	82.98	69.79
FR-148	77.78	76.25	71.01	56.62	55.37	70.05
FR-150	96.80	95.91	89.89	85.40	84.29	73.55
FR-178	84.31	83.07	76.05	71.60	65.16	78.76
FR-179	82.43	81.51	74.34	65.90	63.16	75.64
FR-180	90.88	89.82	85.89	83.00	82.71	80.10

**Table 2 biomimetics-08-00541-t002:** Average $accu_{y}$ analysis results of the BGRODL-OC approach on test 004 and 007 datasets.

Average Accuracy (%)						
Methods	BGRODL-OC	CBODL-RPD	DLADT-PW	RS-CNN	FR-CNN	MDT
Test-004	99.32	99.06	98.35	97.90	87.42	85.17
Test-007	93.18	92.27	89.50	84.78	80.51	74.61

**Table 3 biomimetics-08-00541-t003:** Comparative TPR outcomes of the BGRODL-OC technique and other existing methods on test sequence 004.

True Positive Rate (TPR) (Test Sequence-004)						
False Positive Rate (FPR)	BGRODL-OC	CBODL-RPD	DLADT-PW	RS-CNN	FR-CNN	MDT
0.00	0.0000	0.0000	0.0000	0.0000	0.0000	0.0000
0.05	0.2194	0.2156	0.1921	0.1920	0.1547	0.1211
0.10	0.5234	0.3685	0.3637	0.3164	0.2785	0.2272
0.15	0.6310	0.4604	0.4927	0.4474	0.4243	0.3380
0.20	0.7702	0.5961	0.6093	0.5462	0.4972	0.5149
0.25	0.9508	0.7120	0.7166	0.6726	0.6170	0.6559
0.30	0.9910	0.8783	0.8579	0.8786	0.8106	0.8008
0.35	0.9845	0.9373	0.9019	0.9388	0.8790	0.8631
0.40	0.9845	0.9113	0.8760	0.9113	0.8507	0.8381
0.45	0.9931	0.9315	0.9340	0.9795	0.8798	0.9278
0.50	0.9998	0.9592	0.9517	0.9998	0.9142	0.9693
0.55	0.9998	0.9667	0.9719	0.9998	0.9466	0.9899
0.60	0.9998	0.9667	0.9869	0.9998	0.9796	0.9950
0.65	0.9998	0.9667	0.9844	0.9998	0.9781	0.9939
0.70	0.9998	0.9794	0.9920	0.9998	0.9886	0.9863
0.75	0.9998	0.9869	0.9931	0.9998	0.9956	0.9941
0.80	0.9998	0.9895	0.9941	0.9998	0.9995	0.9940
0.85	0.9998	0.9931	0.9931	0.9998	0.9986	0.9939
0.90	0.9998	0.9956	0.9941	0.9998	0.9994	0.9958
0.95	0.9998	0.9956	0.9931	0.9998	0.9999	0.9939
1.00	0.9998	0.9956	0.9931	0.9997	0.9994	0.9950

**Table 4 biomimetics-08-00541-t004:** Comparative TPR outcomes of the BGRODL-OC technique and other existing methods on test sequence 007.

True Positive Rate (Test Sequence-007)						
False Positive Rate	BGRODL-OC	CBODL-RPD	DLADT-PW	RS-CNN	FR-CNN	MDT
0.00	0.0000	0.0000	0.0000	0.0000	0.0000	0.0000
0.05	0.6228	0.4448	0.3607	0.222	0.3609	0.3034
0.10	0.6303	0.4703	0.4006	0.3995	0.4312	0.3429
0.15	0.8952	0.7162	0.5475	0.5133	0.632	0.455
0.20	0.9586	0.7936	0.5991	0.7132	0.7568	0.5312
0.25	0.9785	0.8235	0.849	0.7872	0.7784	0.6676
0.30	0.956	0.8952	0.919	0.8625	0.8965	0.788
0.35	0.9257	0.9583	0.9499	0.9677	0.8957	0.8363
0.40	0.926	0.9653	0.9539	0.9597	0.9317	0.999
0.45	0.9436	0.8146	0.8561	0.8397	0.8626	0.831
0.50	0.9711	0.8635	0.8791	0.8757	0.9334	0.8591
0.55	0.9711	0.8818	0.8911	0.8923	0.9366	0.8576
0.60	0.9711	0.9028	0.9136	0.8935	0.9629	0.8584
0.65	0.9711	0.9029	0.9225	0.9051	0.9617	0.8771
0.70	0.9711	0.9077	0.9567	0.9113	0.9625	0.9147
0.75	0.9711	0.9343	0.9827	0.9264	0.9603	0.9397
0.80	0.9971	0.9468	0.9822	0.9427	0.961	0.9407
0.85	0.9971	0.9464	0.9867	0.9675	0.9771	0.9406
0.90	0.9971	0.9571	0.9798	0.9908	0.968	0.9772
0.95	0.9971	0.9932	0.9691	0.9967	0.9602	0.9701
1.00	0.9971	0.9940	0.9981	0.9967	0.9857	0.9701

**Table 5 biomimetics-08-00541-t005:** AUC score and CT analysis outcomes of the BGRODL-OC technique and other algorithms.

Methods	AUC Score (%)	Computational Time (s)
BGRODL-OC	97.80	1.08
CBODL-RPD	96.54	2.90
DLADT-PW	89.24	2.75
RS-CNN	90.03	3.19
FR-CNN	89.88	2.90
MDT	89.28	3.56

## Data Availability

No new data were created or analyzed in this study. Data sharing is not applicable to this article.

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
