# Peer review of "Bioinspired Garra Rufa Optimization-Assisted Deep Learning Model for Object Classification on Pedestrian Walkways"

_biomimetics, 2023, doi:10.3390/biomimetics8070541_

Round 1

Reviewer 1 Report

Comments and Suggestions for Authors

This research discusses the  Bioinspired Garra Rufa optimization Assisted Deep Learning Model for Object Classification on Pedestrian Walkways. The study provides insight into the design of the Bioinspired Garra Rufa optimization Assisted Deep Learning Model for Object Classification (BGRODL-OC) technique on Pedestrian Walkways. The objective of the BGRODL-OC technique is to recognize the presence of pedestrians and other objects in the surveillance video. I have several comments for authors to improve their paper as follows:

1. It would be good if the Authors would describe the novel cases drawn from prior cases that they used as research contributions.

2. It is desired to discuss the model complexity of the proposed model and the baseline models. This would show their efficiency in terms of runtime and space complexity.

3. I suggest the authors give a problem statement or use case in Methodology so as to help readers understand how to make a recommendation based on the dataset.

4. I suggest the authors show the detailed process of parameters. However, the proposed method shows good performance. I also suggest the authors to adding a part of applications to discuss the accurately identifying tomato leaf disease types and reducing crop losses based the proposed method and obtained results.

5. Motivation of the paper is not clearly mentioned  and not clearly mention the need for paper at this moment as few better technologies is already exist.

Author Response

Dear Editor, 

Please find the revised version of our manuscript.
Thank You.

Reviewer 2 Report

Comments and Suggestions for Authors

In general, the technical structure of the paper is well done. The paper is well supported by mathematics.

However,

1.       Some references are needed for past work (not only the last 3 or 4 years) in the introduction to give the progress that has been noted until today so that to be able for the reader to see the contribution of their work in this field (state of the art).

2.       The authors should read the manuscript carefully and rewrite the abbreviations after the first appearance of their meaning. It is better in the abstract to give only the full name not the abbreviation. Some of the abbreviations are not given in the full meaning in their first used i.e. AUC, TRP.

3.       The authors should state the novelty of their work clearly and what are the advantages and disadvantages of their work from others, both in the abstract and in the introduction.

4.       The authors should give more explanations in the experimental results. So section 4, will be extended and more clear to the readers.

  Finally,

5.       The authors should give in the conclusions extended highlights of their work and highlights for future research.

Some mistakes at the Tables and Figures: what does it means i.e. Accurac or Accuy

Comments on the Quality of English Language

Minor editing of English language required

Author Response

(The authors gave the same response as above.)

Round 2

Reviewer 1 Report

Comments and Suggestions for Authors

Can be accepted in the current version